# Federated Learning under Partially Class-Disjoint Data via Manifold Reshaping

**Ziqing Fan, Jiangchao Yao$^{\boxtimes}$, Ruipeng Zhang**        *{zqfan_knight, sunarker, zhangrp}@sjtu.edu.cn*
*Cooperative Medianet Innovation Center, Shanghai Jiao Tong University*
*Shanghai AI Laboratory*

**Lingjuan Lyu**        *Lingjuan.Lv@sony.com*
*Sony AI*

**Ya Zhang, Yanfeng Wang$^{\boxtimes}$**        *{ya_zhang, wangyanfeng}@sjtu.edu.cn*
*Cooperative Medianet Innovation Center, Shanghai Jiao Tong University*
*Shanghai AI Laboratory*

**Reviewed on OpenReview:** *https://openreview.net/forum?id=jLJTqJXAG7*

## Abstract

Statistical heterogeneity severely limits the performance of federated learning (FL), motivating several explorations *e.g.,* FedProx, MOON and FedDyn, to alleviate this problem. Despite effectiveness, their considered scenario generally requires samples from almost all classes during the local training of each client, although some covariate shifts may exist among clients. In fact, the natural case of partially class-disjoint data (PCDD), where each client contributes a few classes (instead of all classes) of samples, is practical yet underexplored. Specifically, the unique collapse and invasion characteristics of PCDD can induce the biased optimization direction in local training, which prevents the efficiency of federated learning. To address this dilemma, we propose a manifold reshaping approach called FedMR to calibrate the feature space of local training. Our FedMR adds two interplaying losses to the vanilla federated learning: one is intra-class loss to decorrelate feature dimensions for anti-collapse; and the other one is inter-class loss to guarantee the proper margin among categories in the feature expansion. We conduct extensive experiments on a range of datasets to demonstrate that our FedMR achieves much higher accuracy and better communication efficiency. Source code is available at: https://github.com/MediaBrain-SJTU/FedMR.

## 1 Introduction

Federated learning (McMahan et al. (2017); Li et al. (2020a); Yang et al. (2019)) has drawn considerable attention due to the increasing requirements on data protection (Shokri & Shmatikov (2015); Zhu & Han (2020); Hu et al. (2021); Li et al. (2021c); Lyu et al. (2020)) in real-world applications like medical image analysis (Guo et al. (2021); Park et al. (2021); Yin et al. (2022); Dou et al. (2021); Jiang et al. (2022)) and autonomous driving (Liang et al. (2019); Pokhrel & Choi (2020)). Nevertheless, the resulting challenge of data heterogeneity severely limits the application of machine learning algorithms (Zhao et al. (2018)) in federated learning. This motivations a plenty of explorations to address the statistical heterogeneity issue and improve the efficiency (Kairouz et al. (2021); Wang et al. (2020); Li et al. (2022)).

Existing approaches to address the statistical heterogeneity can be roughly summarized into two categories. One line of research is to constrain the parameter update in local clients or in the central server. For example, FedProx (Li et al. (2020b)), FedDyn (Acar et al. (2020)) and FedDC (Gao et al. (2022)) explore how to reduce the variance or calibrate the optimization by adding the proximal regularization on parameters in FedAvg (McMahan et al. (2017)). The other line of research focuses on constraining the representation

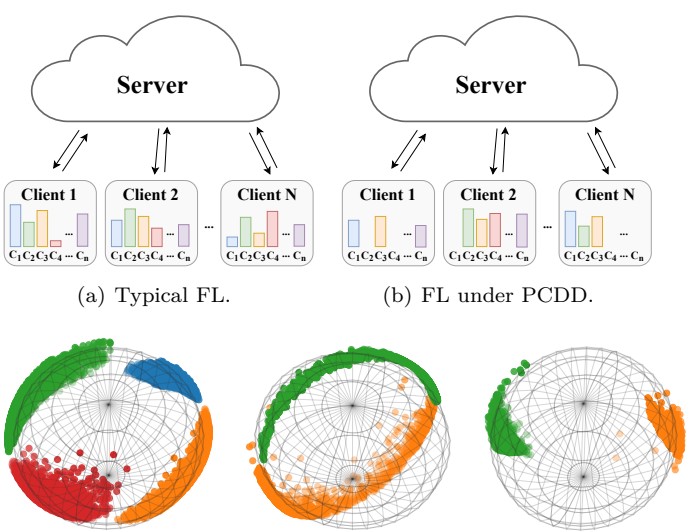

(a) Typical FL.      (b) FL under PCDD.

(c) Global / Collapsed / Reshaped (by FedMR) feature space.

Figure 1: Federated learning under partially class-disjoint data (PCDD).

from the model to implicitly affect the update. FedProc (Mu et al. (2021)) and FedProto (Tan et al. (2022)) introduce prototype learning to help local training, and MOON (Li et al. (2021b)) utilizes contrastive learning to minimize the distance between representations learned by local model and global model, and maximize the distance between representations learned by local model and previous local model. However, all these methods validate their efficiency mostly under the support from all classes of samples in each client while lacking a well justification on a natural scenario, namely *partially class-disjoint data w.r.t.* classes.

As illustrated in Figure 1(a), in typical federated learning, each client usually contains all classes of samples but under different covariate shifts, and all clients work together to train a global model. However, in the case of PCDD (Figure 1(b)), there are only a small subset of categories in each client and all clients together provide information of all classes. Such a situation is very common in real-world applications. For example, there are shared and distinct Thyroid diseases in different hospitals due to regional diversity (Gaitan et al. (1991)). Hospitals from different regions can construct a federation to learn a comprehensive model for the diagnostic of Thyroid diseases but suffer from the PCDD challenge. We conduct a toy study on a simulated dataset (see details in the Appendix B), and visualize the feature space under centralized training (the left panel of Figure 1(c)) and local training under PCDD (the middle panel of Figure 1(c)) by projecting on the unit sphere. As shown in Figure 1(c), PCDD induces a dimensional collapse onto a narrow area due to the lack of support from all classes, and causes a space invasion to the missing classes. Previous approaches such as FedProc and MOON may implicitly constrain the space invasion by utilizing class prototypes or global features generated from the global model, and methods like FedProx, FedDyn, and FedNova could also help a bit from the view of optimization. However, these methods are not oriented towards the PCDD issue, and they are inefficient to avoid the collapse and invasion characteristics of PCDD and achieve sub-optimal performance from the both view of experimental performance shown in Table 1 and the feature variance of all methods shown in Table 7.

To address this dilemma, we propose a manifold-reshaping approach called FedMR to properly prevent the degeneration caused by the locally class missing. FedMR introduces two interplaying losses: one is intra-class loss to decorrelate feature space for anti-collapse; and another one is the inter-class loss to guarantee the proper margin among categories by means of global class prototypes. The right panel of Figure 1(c) provides a rough visualization of FedMR. Theoretically, we analyze the benefit from the interaction of the intra-class loss and the inter-class loss under PCDD, and empirically, we verify the effectiveness of FedMR compared with the current state-of-the-art methods. Our contributions can be summarized as follows:

- We are among the first attempts to study dimensional collapse and space invasion challenges caused by PCDD in Generic FL that degenerates embedding space and thus limits the model performance.

- We introduce a approach termed as FedMR, which decorrelates the feature space to avoid dimensional collapse and constructs a proper inter-class margin to prevent space invasion. Our theoretical analysis confirms the rationality of the designed losses and their benefits to address the dilemma of PCDD.

- We conduct a range of experiments on multiple benchmark datasets under PCDD and a real-world disease dataset to demonstrate the advantages of FedMR over the state-of-the-art methods. We also develop several variants of FedMR to consider the communication cost and privacy concerns.

## 2    Related Works

### 2.1   Federated Learning

There are extensive works to address the *statistical heterogeneity* in federated learning, which induces the bias of local training due to the covariate shifts among clients (Zhao et al. (2018); Li et al. (2022); Zhang et al. (2023a)). A line of research handles this problem by adding constraints like normalization or regularization on model weights in the local training or in the server aggregation. FedProx (Li et al. (2020b)) utilizes a proximal term to limit the local updates so as to reduce the bias, and FedNova (Wang et al. (2020)) introduces the normalization on total gradients to eliminate the objective inconsistency. FedDyn (Acar et al. (2020)) makes the global model and local models approximately aligned in the limit by proposing a dynamic regularizer for each client at each round. FedDC (Gao et al. (2022)) reduces the inconsistent optimization on the client-side by local drift decoupling and correction. Another line of research focuses on constraining representations from local models and the global model. MOON (Li et al. (2021b)) corrects gradients in local training by contrasting the representation from local model and that from global model. FedProc (Mu et al. (2021)) utilizes prototypes as global information to help correct local representations. Our FedMR is also conducted on representations of samples and classes but follows a totally different problem and spirit.

### 2.2   Representation Learning

The collapse problem is also an inevitable concern in the area of representation learning. In their research lines, there are several attempts to prevent the potential collapse issues. For example, in self-supervised learning, Barlow Twins, VICReg and Shuffled-DBN (Bardes et al. (2022); Zbontar et al. (2021); Hua et al. (2021)) manipulate the rank of the (co-)variance matrix to prevent the potential collapse. In contrastive learning, DirectCLR (Jing et al. (2021)) directly optimizes the representation space without an explicit trainable classifier to promote a larger feature diversity. In incremental learning, CwD (Shi et al. (2022)) applies a similar technique to prevent dimensional collapse in the initial phase. In our PCDD case, it is more challenging, since we should not only avoid collapse but also avoid the space invasion in the feature space.

### 2.3   Federated Prototype Learning

In many vision tasks (Snell et al. (2017); Yang et al. (2018); Deng et al. (2021)), prototypes are the mean values of representations of a class and contain information like feature structures and relationships of different classes. Since prototypes are population-level statistics of features instead of raw features, which are relatively safe to share, prototype learning thus has been applied in federated learning. In Generic FL, FedProc (Mu et al. (2021)) utilizes the class prototypes as global knowledge to help correct local training. In Personalized FL, FedProto (Tan et al. (2022)) shares prototypes instead of local gradients to reduce communication costs. We also draw spirits from prototype learning to handle PCDD. Besides, to make fair comparison to methods without prototypes and better reduce extreme privacy concerns, we conduct a range of auxiliary experiments in Section 4.4.

## 3    The Proposed Method

### 3.1   Preliminary

**PCDD Definition.**    There are many nonnegligible real-world PCDD scenarios. ISIC2019 dataset (Codella et al. (2018); Tschandl et al. (2018); Combalia et al. (2019)), a region-driven *subset* of types of Thyroid

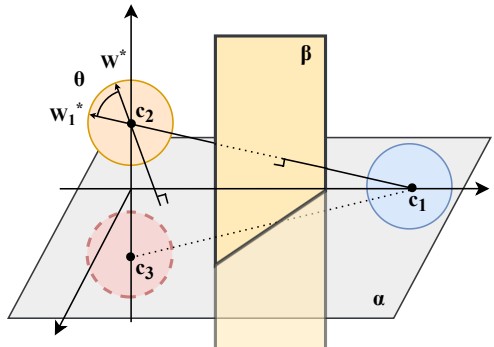

Figure 2: An illustration about the shift of the optimization direction under PCDD. Here, we assume our client contains two classes $c_1$ and $c_2$ with one missing class $c_3$. $\mathbf{w}^*$ is the optimal classifier direction for $c_2$ (perpendicular to the plane $\alpha$) when all classes exist, and $\mathbf{w}_1^*$ is the learned classifier direction when $c_3$ is missing, which can be inferred by the decision plane $\beta$ between $c_1$ and $c_2$. As can be seen, PCDD leads to the angle shift $\theta$ in the optimization.

diseases in the hospital systems, is utilized in our experiments. In landmark detection (Weyand et al. (2020)) for thousands of categories with data locally preserved, most contributors only have a *subset* of categories of landmark photos where they live or traveled before, which is also a scenario for the federated PCDD problem. To make it clear, we first define some notations of the partially class-disjoint data situation in federated learning. Let $C$ denote the collection of full classes and P denote the set of all local clients. Considering the real-world constraints like privacy or environmental limitation, each local client may only own the samples of partial classes. Thus, for the $k$-th client $P_k$, its corresponding local dataset $D_k$ can be expressed as $D_k = \{(x_{k,i}, y_{k,i}) | y_{k,i} = c \in C_k\}$, where $C_k \subsetneq C$. The number of samples of the class $c$ ($c \in C_k$) in $P_k$ is $N_k^c$. We denote a local model $f(\cdot; \mathbf{w}_k)$ on all clients as two parts: a backbone network $f_1(\cdot; \mathbf{w}_{k,1})$ and a linear classifier $f_2(\cdot; \mathbf{w}_{k,2})$. The loss of the $k$-th client can be formulated as

$$\ell_k^{\text{cls}}(D_k; w_k) = \frac{1}{N_k} \sum_{i=1}^{N_k} \ell(y_{k,i}, f_2(z_{k,i}; w_{k,2}))\big|_{z_{k,i}=f_1(x_{k,i}; w_{k,1})},$$

where $z_{k,i}$ is the feature representation of the input $x_{k,i}$ and $\ell(\cdot, \cdot)$ is the loss measure. Under PCDD, we can emprically find the dimensional collapse and the space invasion problems about representation.

**FedAvg.** The vanilla federated learning via FedAvg consists of four steps (McMahan et al. (2017)): 1) In round $t$, the server distributes the global model $\mathbf{w}^t$ to clients that participate in the training; 2) Each local client receives the model and continues to train the model, *e.g.,* the $k$-th client conducts the following,

$$\mathbf{w}_k^t \leftarrow \mathbf{w}_k^t - \eta \nabla \ell_k(b_k^t; \mathbf{w}_k^t), \tag{1}$$

where $\eta$ is the learning rate, and $b_k^t$ is a mini-batch of training data sampled from the local dataset $\mathcal{D}_k$. After $E$ epochs, we acquire a new local model $\mathbf{w}_k^t$; 3) The updated models are then collected to the server as $\{\mathbf{w}_1^t, \mathbf{w}_2^t, \ldots, \mathbf{w}_K^t\}$; 4) The server performs the following aggregation to acquire a new global model $\mathbf{w}^{t+1}$,

$$\mathbf{w}^{t+1} \leftarrow \sum_{k=1}^{K} p_k \mathbf{w}_k^t, \tag{2}$$

where $p_k$ is the proportion of sample number of the $k$-th client to the sample number of all the participants, *i.e.,* $p_k = N_k / \sum_{k'=1}^{K} N_{k'}$. When the maximal round $T$ reaches, we will have the final optimized model $\mathbf{w}^T$.

## 3.2 Motivation

In Figure 2, we illustrate a low-dimensional example to characterize the directional shift of the *local* training under PCDD on the client side. In the following, we use the parameter aggregation of a linear classification

to study the directional shift of *global* model in the server, which further clarifies the adverse effect of PCDD. Similar to Figure 2, let $c_1$, $c_2$ and $c_3$ denote three classes of samples on a circular face respectively centered at $(1, 0)$, $(0, \sqrt{3})$ and $(0, -\sqrt{3})$ with radius r$=\frac{1}{2}$ under a uniform distribution. Then, if there are samples of all classes in each local client, we will get the optimal weight for all categories as follows:

$$w^* = \begin{bmatrix} 1 & 0 \\ -\frac{\sqrt{3}}{2} & \frac{1}{2} \\ -\frac{\sqrt{3}}{2} & -\frac{1}{2} \end{bmatrix}.$$

Note that, we omit the bias term in linear classification for simplicity. Conversely, among total three participants, if each participant only has the samples of two classes, e.g., $(c_1, c_2)$, $(c_1, c_3)$ and $(c_2, c_3)$ respectively, then their learned weights can be inferred as follows:

$$w_1^*, \, w_2^*, \, w_3^* = \begin{bmatrix} \frac{1}{2} & -\frac{\sqrt{3}}{2} \\ -\frac{1}{2} & \frac{\sqrt{3}}{2} \\ 0 & 0 \end{bmatrix}, \begin{bmatrix} \frac{1}{2} & \frac{\sqrt{3}}{2} \\ 0 & 0 \\ -\frac{1}{2} & -\frac{\sqrt{3}}{2} \end{bmatrix}, \begin{bmatrix} 0 & 0 \\ 0 & 1 \\ 0 & -1 \end{bmatrix}.$$

After the server aggregation, we have the estimated weight

$$\hat{w}^* = \begin{bmatrix} \frac{1}{3} & 0 \\ -\frac{1}{6} & \frac{\sqrt{3}+2}{6} \\ -\frac{1}{6} & -\frac{\sqrt{3}+2}{6} \end{bmatrix}.$$

Then, we can find that except the difference on amplitude between $w^*$ and $\hat{w}^*$, a more important issue is the optimization direction for $c_2$ (or $c_3$) shifts about $45°$ by computing the angle between vector $(-\frac{\sqrt{3}}{2}, \frac{1}{2})$ and vector $(-\frac{1}{6}, \frac{\sqrt{3}+2}{6})$ (or between vector $(-\frac{\sqrt{3}}{2}, -\frac{1}{2})$ and vector $(-\frac{1}{6}, -\frac{\sqrt{3}+2}{6})$). Actually, the angle shift can be enlarged in some real-world applications, when the hard negative classes are missing. However, if we can have the statistical centroid of the locally missing class in the local client, namely $c_3$ in the case of Figure 2, it is easy to find that the inferred optimal $\hat{w}^*$ is same to $w^*$ as the decision plane can be normally characterized with the support of the single point $c_3$. This inspires us to design the subsequent method[1].

### 3.3 Manifold Reshaping

As the aforementioned analysis, PCDD in federated learning leads to the directional shift of optimization both in the local models and in the global model. An empirical explanation is that the feature representation of the specific class that should support classification is totally missing, inducing the feature representation of other observed classes arbitrarily distributes as a greedy collapsed manifold, as shown in Figure 1(c). To address this problem, we explore a manifold-reshaping method from both the intra-class perspective and the inter-class perspective. In the following, we will present two interplaying losses and our framework.

#### 3.3.1 Intra-Class Loss

The general way to prevent the representation from collapsing into a low-dimensional manifold, is to decorrelate dimensions for different patterns and expand the intrinsic dimensionality of each category. Such a goal can be implemented by manipulating the rank of the covariance matrix regarding representation. Specifically, for each client, we can first compute the class-level normalization for the representation $z_{k,i}^c \in \mathbb{R}^d$ as $\hat{z}_{k,i}^c = \frac{z_{k,i}^c - \mu_k^c}{\sigma_k^c}$, where $\mu_k^c$ and $\sigma_k^c$ are the mean and standard deviation of features belonging to class $c$ and calculated as: $\mu_k^c = \frac{1}{n_{k,c}} \sum_{i=1}^{n_{k,c}} z_{k,i}^c$ and $\sigma_k^c = \sqrt{\frac{1}{n_{k,c}} \sum_{i=1}^{n_{k,c}} (z_{k,i}^c - \mu_k^c)^2}$. Then, we compute an intra-class covariance matrix based on the above normalization for each observed class in the $k$-th client:

$$M_k^c = \frac{1}{N_k^c - 1} \sum_{i=1}^{N_k^c} \left( \hat{z}_{k,i}^c \left( \hat{z}_{k,i}^c \right)^\top \right).$$

---

[1]Note that, we would like to point out that in the real-world case, we cannot acquire such statistical centroid in advance but have to resort to the training process along with the special technique design.

Since each eigenvalue of $M_k^c \in \mathbb{R}^{d \times d}$ characterizes the importance of a feature dimension within class, we can make them distributed uniformly to prevent the dimensional collapse of each observed class. However, considering the learnable pursuit of machine learning algorithms, we actually cannot directly optimize eigenvalues to reach this goal. Fortunately, it is possible to use an equivalent objective as an alternative, which is clarified by the following lemma.

**Lemma 1.** *Assuming a covariance matrix $M \in \mathbf{R}^{d \times d}$ computed from the feature of each sample with the standard normalization, and its eigenvalues $\{\lambda_1, \lambda_2, ..., \lambda_d\}$, we will have the following equality that satisfied*

$$\sum_{i=1}^{d} (\lambda_i - \frac{1}{d} \sum_{j=1}^{d} \lambda_j)^2 = ||M||_F^2 - d.$$

The complete proof is summarized in the Appendix A.2. From Lemma 1, we can see that pursuing the uniformity of the eigenvalues for the covariance matrix can transform into minimizing the Frobenius norm of the covariance matrix. Therefore, our intra-class loss to prevent the undesired dimensional collapse for observed classes is formulated as

$$\ell_k^{\text{intra}} = \frac{1}{|C_k|} \sum_{c \in C_k} ||M_k^c||_F^2. \tag{3}$$

### 3.3.2 Inter-Class Loss

Although the intra-class loss helps decorrelate the feature dimensions to prevent collapse, the resulting space invasion for the missing classes can be concomitantly exacerbated. Thus, it is important to guarantee the proper space of the missing classes in the expansion as encouraged by equation 3. To address this problem, we maintain a series of global class prototypes and transmit them to local clients as support of the missing classes in the feature space. Concretely, we first compute the class prototypes in the $k$-th client as the average of feature representations (Snell et al. (2017); Yang et al. (2018); Deng et al. (2021)):

$$\left\{ g_k^c \middle| g_k^c \leftarrow \frac{1}{N_k^c} \sum_{i=1}^{N_k^c} z_{k,i}^c \text{ for } c \in C_k \right\}.$$

Then, all client-level prototypes are submitted to the server along with local models in federated learning. In the central server, the global prototypes for all classes are updated as

$$\left\{ g_c^t \middle| g_c^t \leftarrow \sum_{k=1}^{K} p_k^c g_k^c \text{ for } c \in C \right\},$$

where $g_c^t$ is the global prototype of the $c$-th class in round $t$ and $p_k^c = N_k^c / \sum_{k=1}^{K} N_k^c$. In the next round, the central server distributes the global prototypes to all clients as the references to avoid the space invasion. Formally, we construct the following margin loss by contrasting the distances from prototypes to the representation of the sample.

$$\ell_k^{\text{inter}} = \frac{1}{|C_k|(|C_k|-1)} \sum_{c_i \in C_k} \sum_{c_j \in C_k \setminus c_i} D_{c_i, c_j}, \tag{4}$$

where $D_{c_i, c_j}$ is defined as:

$$D_{c_i, c_j} = \frac{1}{N_k^c} \sum_{n=1}^{N_k^c} \max\{||z_{k,n}^{c_i} - g_{c_i}^t|| - ||z_{k,n}^{c_i} - g_{c_j}^t)||, 0\}.$$

In the following, we use a theorem to show how inter-class loss jointly with intra-class loss makes the representation of the missing classes approach to the optimal.

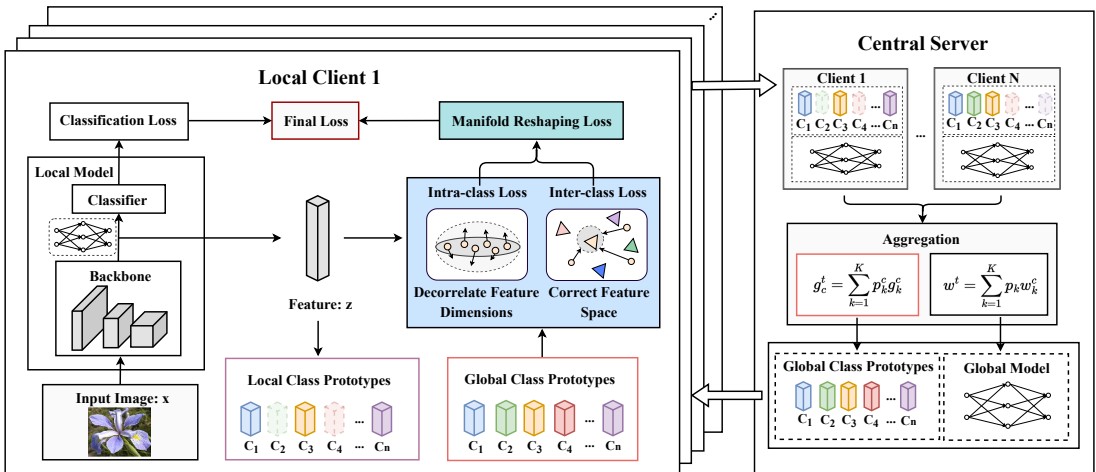

Figure 3: The framework of FedMR. On the client side, except the vanilla training with the classification loss, the manifold-reshaping parts, *i.e.,* the intra-class loss and the inter-class loss, respectively help conduct the feature decorrelation to avoid the dimensional collapse, and leverage the global prototypes to construct the proper margin among classes to prevent the space invasion. On the server side, except the model aggregation, the global class prototypes are also the reference for missing classes participating in the local training.

---

**Algorithm 1** FedMR

---

**Input**: a set of $K$ clients that participate in each round, the initial model weights $\mathbf{w}^0$, the maximal round $T$, the learning rate $\eta$, the local training epochs $E$.

    **for** $t = 0, 1, \ldots, T-1$ **do**

        randomly sample $K$ clients

        updates global model weights and global class prototypes ($\mathbf{w}^t \leftarrow \sum_{k=1}^{K} p_k^t \mathbf{w}_k^{t-1} \ \forall c, g_c^t \leftarrow \sum_{k=1}^{K} p_k^c g_k^c$).

        distribute $\mathbf{w}^t$ and $G_c\{g_1^t, g_2^t, ..., g_C^t\}$ to the $K$ clients.

        **do in parallel for** $\forall k \in K$ **clients**

            $\mathbf{w}_k^t \leftarrow \mathbf{w}^t$.

            **for** $\tau = 0, 1, ..., E-1$ **do**

                sample a mini-batch from local dataset and perform updates($\mathbf{w}_k^t \leftarrow \mathbf{w}_k^t - \eta \nabla \mathcal{L}_k(b_k^t, G_c; \mathbf{w}_k^t)$ ).

            **end for**

            update local class prototypes ($\forall c, \ g_k^c \leftarrow \sum_{i=1}^{n_k^c} \frac{1}{n_k^c} z_i^c$), and submit $\mathbf{w}_k^t$ and $\{g_k^1, g_k^2, ..., g_k^C\}$ to server.

        **end in parallel**

    **end for**

---

**Theorem 1.** *Let the global optimal representation for class $c$ be $g_c^* = [a_{c,1}^*, ..., a_{c,d}^*]$, and $z_k^{c,t}$ be the representation of sample $x$ in the class $c$ of the $k$-th client. Assuming that $\forall i$, both $|a_{c,i}^*|$ and $z_{k,i}^{c,t}$ are upper bounded by $G$, and all dimensions are disentangled, in round $t$, the $i$-th dimension of local representation $z_k^{c,t}$ satisfies*

$$|z_{k,i}^{c,t} - a_{c,i}^*| \leq 2(1 - \hat{p}_k^c \Gamma)G + \delta\Gamma,$$

*where $\hat{p}_k^c$ is the accumulation regarding the $i$-th dimension of the class-$c$ prototype, $\Gamma = \frac{1-(p_k^c)^t}{1-p_k^c}$, $(p_k^c)^t$ refers to the $p_k^c$ raised to the power of $t$, and $\delta$ is the maximum margin of the inter-loss term.*

Note that, Theorem 1 shows five critical points: 1) The proof of the theorem requires each dimension of the representation to be irrelevant to each other, which is achieved by the intra-class loss. Although disentanglement of dimensions might not be totally achieved in practical, empirically, we find that the intra-class loss converges and maintains a relatively low value easily, meaning that the model achieves good

decorrelation. In the Appendix, we show the training curve of intra-class loss during the federated training. 2) In the theorem, $\delta$ is a trade-off of training stability and theoretical results determined by the margin. The larger the margin, the larger the $\delta$, however the more stable the local training is. This is because the global prototypes are not very accurate in the early stage of local training and directly minimizing the distance of samples to their global class prototypes can bring side effect to the feature diversity. When the margin is removed ($D_{c_i, c_j} = \frac{1}{N_k^c} \sum_{n=1}^{N_k^c} ||z_{k,n}^{c_i} - g_{c_i}^t||$), $\delta$ will be zero. 3) Without considering $\delta$, as $t$ increases, $\Gamma$ is smaller and representation $z_k^c$ is closer to global optimal prototype $g_c^*$, showing the promise of our method. 4) When t is large enough, we can get an upper bound $2\frac{1-p_k^c-\hat{p}_k^c}{1-p_k^c}G$, meaning more clients with the specific dimensional information participating in the training, the tighter the upper bound is. When all other clients can provide the support ($\hat{p}_k^c = 1 - p_k^c$), the error will be 0. 5) While $\hat{p}_k^c$ denotes the proportions of a subset of clients that can provide the support information for this dimension, the theoretical result depends on the class distribution and overlap across the clients. The complete proof is summarized in the Appendix A.1.

### 3.3.3 The Total Framework

After introducing the intra-class loss and the inter-class loss, we give the total framework of FedMR. On the client side, local models are trained on their partially class-disjoint datasets. Through manifold-reshaping loss, dimensions of the representation are decorrelated and local class subspace is corrected to prevent the space invasion, and gradually approach the global space partition. The total local objective including the vanilla classification loss can be written as

$$\mathcal{L}_k = \ell_k^{\text{cls}} + \underbrace{\left(\mu_1 \ell_k^{\text{intra}} + \mu_2 \ell_k^{\text{inter}}\right)}_{\text{manifold reshaping}}, \tag{5}$$

where $\mu_1$ and $\mu_2$ are the balancing hyperparameters and will be discussed in the experimental part. In Figure 3, we illustrate the corresponding structure of FedMR and formulate the training procedure in Algorithm 1. In terms of the privacy concerns about the prototype transmission and the communication cost, we will give the comprehensive analysis on FedMR about these factors.

## 4 Experiment

### 4.1 Experimental Setup

**Datasets.** We adopt four popular benchmark datasets SVHN (Netzer et al. (2011)), FMNIST (Xiao et al. (2017)), CIFAR10 and CIFAR100 (LeCun et al. (1998)) in federated learning and a real-world PCDD medical dataset ISIC2019 (Codella et al. (2018); Tschandl et al. (2018); Combalia et al. (2019)) to conduct experiments. Regarding the data setup, although Dirichelet Distribution is popular to split data in FL, it usually generates diverse imbalance data coupled with occasionally PCDD. In order to better study pure PCDD, for the former four benchmarks, we split each dataset into $\varrho$ clients, each with $\varsigma$ categories, abbreviated as $P\varrho C\varsigma$. For example, P10C10 in CIFAR100 means that we split CIFAR100 into 10 clients, each with 10 classes. Please refer to the detailed explanations and strategies in the Appendix. ISIC2019 is a real-world federated application under the PCDD situation and the data distribution among clients is shown in the Appendix C.1.3. We follow the settings in Flamby benchmark (Terrail et al. (2022)).

**Implementation.** We compare FedMR with FedAvg (McMahan et al. (2017)) and multiple state-of-the-arts including FedProx (Li et al. (2020b)), FedProc (Karimireddy et al. (2020)), FedNova (Li et al. (2021b)), MOON (Wang et al. (2020)), FedDyn (Acar et al. (2020)) and FedDC (Gao et al. (2022)). To make a fair and comprehensive comparison, we utilize the same model for all approaches and three model structures for different datasets: ResNet18 (He et al. (2016)) (follow (Li et al. (2021b; 2022))) for SVHN, FMNIST and CIFAR10, wide ResNet (Zagoruyko & Komodakis (2016)) for CIFAR100 and EfficientNet (Tan & Le (2019)) for ISIC2019. The optimizer is SGD with a learning rate 0.01, the weight decay $10^{-5}$ and momentum 0.9. The batch size is set to 128 and the local updates are set to 10 epochs for all approaches. The detailed information of the model and training parameters are given in the Appendix.

Table 1: Performance of FedMR and a range of state-of-the-art approaches on four datasets under PCDD partitions. Datasets are divided into $\varrho$ clients and each client has $\varsigma$ classes (denoted as $P\varrho C\varsigma$). We compute average accuracy of all partitions, highlight results of FedMR, underline results of best baseline, and show the improvement of FedMR to FedAvg (subscript of FedMR) and to the best baseline ($\Delta$).

| Datasets | Split | FedAvg | FedProx | FedProc | FedNova | MOON | FedDyn | FedDC | FedMR | $\Delta$ |
|---|---|---|---|---|---|---|---|---|---|---|
| FMNIST | P5C2 | 67.29 | 69.60 | 66.28 | 66.87 | 66.82 | 71.01 | 68.50 | $\mathbf{75.51}_{8.22\%\uparrow}$ | +4.50 |
| | P10C2 | 67.33 | 67.76 | 69.08 | 48.44 | 67.93 | 67.16 | 67.36 | $\mathbf{74.97}_{7.64\%\uparrow}$ | +5.89 |
| | P10C3 | 81.67 | 80.93 | 82.06 | 83.20 | 83.42 | 83.00 | 83.24 | $\mathbf{83.55}_{1.88\%\uparrow}$ | +0.13 |
| | P10C5 | 88.53 | 89.22 | 89.23 | 88.86 | 88.98 | 88.38 | 89.22 | $\mathbf{90.04}_{1.51\%\uparrow}$ | +0.81 |
| | IID | 91.93 | 91.95 | 92.06 | 91.84 | 92.12 | 91.76 | 92.15 | $\mathbf{92.19}_{0.26\%\uparrow}$ | +0.04 |
| | avg | 79.35 | 79.89 | 79.74 | 75.84 | 79.87 | 80.26 | 80.09 | $\mathbf{83.45}_{4.10\%\uparrow}$ | +3.19 |
| SVHN | P5C2 | 81.85 | 81.83 | 79.54 | 81.11 | 81.60 | 79.89 | 81.63 | $\mathbf{83.10}_{1.25\%\uparrow}$ | +1.27 |
| | P10C2 | 78.92 | 79.60 | 78.75 | 66.86 | 79.83 | 76.24 | 78.96 | $\mathbf{82.47}_{3.55\%\uparrow}$ | +2.64 |
| | P10C3 | 87.70 | 87.40 | 88.13 | 87.50 | 87.83 | 87.27 | 88.05 | $\mathbf{89.13}_{1.43\%\uparrow}$ | +1.00 |
| | P10C5 | 91.20 | 91.24 | 91.63 | 92.09 | 91.16 | 90.17 | 91.64 | $\mathbf{92.18}_{0.98\%\uparrow}$ | +0.09 |
| | IID | 92.74 | 92.89 | 93.57 | 92.62 | 93.12 | 92.26 | 92.90 | $\mathbf{93.04}_{0.30\%\uparrow}$ | -0.53 |
| | avg | 86.48 | 86.59 | 86.32 | 83.87 | 86.71 | 85.17 | 86.64 | $\mathbf{87.98}_{1.50\%\uparrow}$ | +1.27 |
| CIFAR10 | P5C2 | 67.68 | 68.18 | 69.27 | 67.57 | 66.86 | 69.64 | 69.18 | $\mathbf{74.19}_{6.51\%\uparrow}$ | +4.55 |
| | P10C2 | 67.27 | 71.09 | 67.02 | 57.79 | 67.61 | 67.74 | 67.64 | $\mathbf{73.32}_{2.23\%\uparrow}$ | +2.23 |
| | P10C3 | 77.82 | 77.89 | 77.87 | 77.22 | 78.42 | 77.99 | 77.94 | $\mathbf{82.75}_{4.93\%\uparrow}$ | +4.33 |
| | P10C5 | 88.22 | 88.34 | 88.19 | 88.20 | 88.00 | 88.35 | 88.14 | $\mathbf{89.06}_{0.84\%\uparrow}$ | +0.71 |
| | IID | 91.88 | 92.14 | 92.62 | 92.37 | 92.56 | 92.29 | 92.85 | $\mathbf{93.06}_{1.18\%\uparrow}$ | +0.21 |
| | avg | 78.58 | 79.53 | 78.99 | 76.63 | 78.69 | 79.20 | 79.15 | $\mathbf{82.48}_{3.90\%\uparrow}$ | +2.95 |
| CIFAR100 | P10C10 | 54.31 | 54.79 | 54.69 | 54.45 | 54.98 | 55.94 | 54.73 | $\mathbf{57.27}_{2.96\%\uparrow}$ | +1.33 |
| | P10C20 | 64.81 | 65.37 | 64.98 | 65.79 | 65.75 | 65.02 | 65.21 | $\mathbf{65.81}_{1.00\%\uparrow}$ | +0.02 |
| | P10C30 | 69.35 | 69.75 | 69.64 | 69.55 | 69.51 | 69.84 | 69.38 | $\mathbf{70.24}_{0.89\%\uparrow}$ | +0.40 |
| | P10C50 | 71.28 | 71.35 | 72.13 | 71.25 | 71.54 | 71.25 | 72.11 | $\mathbf{72.17}_{0.89\%\uparrow}$ | +0.04 |
| | IID | 72.28 | 72.55 | 73.07 | 72.66 | 73.01 | 73.04 | 72.77 | $\mathbf{72.79}_{0.51\%\uparrow}$ | -0.28 |
| | avg | 66.41 | 66.76 | 66.90 | 66.74 | 66.96 | 67.22 | 66.84 | $\mathbf{67.66}_{1.25\%\uparrow}$ | +0.44 |

Table 2: Global test accuracy of methods on CIFAR10 and CIFAR100 under larger scale of clients (P10, P50 and P100) and ISIC2019. $P\varrho C\varsigma$ denotes that the dataset is divided into $\varrho$ clients and each client has $\varsigma$ classes of samples. We highlight results of FedMR, underline results of best baseline, and show the improvement of FedMR to FedAvg (bottom right corner of results) and to the best baseline ($\Delta$).

| Datasets | Split | FedAvg | FedProx | FedProc | FedNova | MOON | FedDyn | FedDC | FedMR | $\Delta$ |
|---|---|---|---|---|---|---|---|---|---|---|
| CIFAR10 | P10C3 | 77.82 | 77.89 | 77.87 | 77.22 | 78.42 | 77.99 | 77.94 | $\mathbf{82.75}_{4.93\%\uparrow}$ | +4.33 |
| | P50C3 | 75.46 | 77.46 | 76.27 | 74.12 | 76.51 | 75.52 | 76.03 | $\mathbf{79.58}_{4.12\%\uparrow}$ | +2.12 |
| | P100C3 | 71.65 | 72.37 | 72.44 | 70.46 | 72.46 | 71.85 | 73.95 | $\mathbf{76.93}_{5.28\%\uparrow}$ | +2.98 |
| CIFAR100 | P10C10 | 54.31 | 54.79 | 54.69 | 54.45 | 54.98 | 55.94 | 54.73 | $\mathbf{57.27}_{2.96\%\uparrow}$ | +1.33 |
| | P50C10 | 49.84 | 51.17 | 51.94 | 50.22 | 52.19 | 50.53 | 51.17 | $\mathbf{53.36}_{3.52\%\uparrow}$ | +1.17 |
| | P100C10 | 47.90 | 48.26 | 49.01 | 48.07 | 48.94 | 49.24 | 48.76 | $\mathbf{49.60}_{1.70\%\uparrow}$ | +0.36 |
| ISIC2019 | Real | 73.14 | 75.41 | 75.26 | 73.62 | 75.46 | 75.07 | 75.25 | $\mathbf{76.55}_{3.41\%\uparrow}$ | +1.09 |

## 4.2 Performance under PCDD

In this part, we compare FedMR with FedAvg and other methods on FMNIST, SVHN, CIFAR10 and CIFAR100 datasets under partially class-disjoint situation. Note that, in FedProx, MOON, FedDyn, FedProc, FedDC and our method, there are parameters that need to set. We use grid search to choose the best parameters for each method. See more concrete settings in the Appendix C.2 and Section 4.1.

As shown in Table 1, with the decreasing class number in local clients, the performance of FedAvg and all other methods greatly drops. However, comparing with all approaches, our method FedMR achieves far better improvement to FedAvg, especially 7.64% improvement *vs.* 1.75% of FedProc for FEMNIST (P10C2) and 6.51% improvement *vs.* 1.96% of FedDyn for CIFAR10 (P5C2). Besides, FedMR also performs better under less PCDD, and on average of all partitions listed in the table, our method outperforms the best baseline by 3.19% on FMNIST and 2.95% on CIFAR10.

Table 3: The communication cost of approaches with prototypes (*i.e.,* FedProc and FedMR) and without prototypes (*i.e.,* FedAvg, FedProx, FedNova, MOON, FedDyn and FedDC).

| Method Type | FMNIST | SVHN | CIFAR10 | CIFAR100 | ISIC2019 |
|---|---|---|---|---|---|
| w/o Prototypes | $11.182M$ | $11.184M$ | $11.184M$ | $36.565M$ | $4.875M$ |
| w/ Prototypes | $11.187M$ | $11.189M$ | $11.189M$ | $36.629M$ | $4.880M$ |
| Additional cost | $0.044\%\uparrow$ | $0.044\%\uparrow$ | $0.044\%\uparrow$ | $0.175\%\uparrow$ | $0.102\%\uparrow$ |

Table 4: Number of communication rounds and the speedup of communication when reaching the best accuracy of FedAvg in federated learning on CIFAR100 under three partition strategies.

| Method | P10C10 | | P50C10 | | P100C10 | |
|---|---|---|---|---|---|---|
| (CIFAR100) | Commu. | Speedup | Commu. | Speedup | Commu. | Speedup |
| FedAvg | 400 | $1\times$ | 400 | $1\times$ | 400 | $1\times$ |
| FedProx | 352 | $1.14\times$ | 370 | $1.08\times$ | 384 | $1.04\times$ |
| FedProc | 357 | $1.12\times$ | 381 | $1.05\times$ | 389 | $1.03\times$ |
| FedNova | 290 | $1.38\times$ | 385 | $1.04\times$ | 382 | $1.05\times$ |
| MOON | 332 | $1.20\times$ | 373 | $1.07\times$ | 395 | $1.01\times$ |
| FedDyn | 178 | $2.25\times$ | 366 | $1.09\times$ | 384 | $1.04\times$ |
| FedDC | 275 | $1.45\times$ | 393 | $1.02\times$ | 387 | $1.03\times$ |
| **FedMR** | **149** | **2.68×** | **293** | **1.37×** | **368** | **1.09×** |

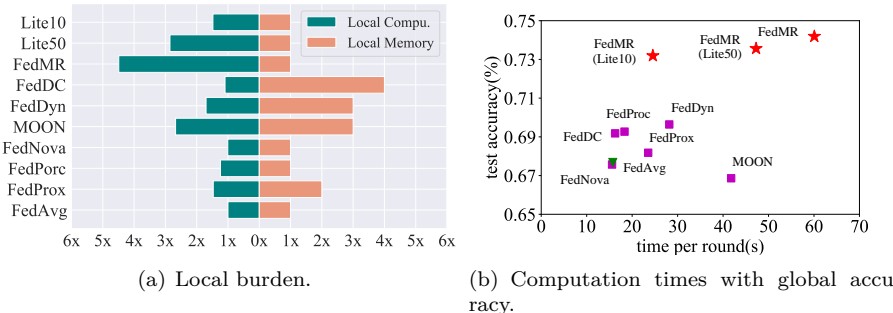

(a) Local burden.

(b) Computation times with global accuracy.

Figure 4: The average memory consuming, computation time of local training and performance on all datasets of all baselines, FedMR and its light versions (Lite 10 and Lite 50) for accelerating.

### 4.3 Scalability and Robustness

In the previous section, we validate the FedMR under 5 or 10 local clients under partially class-disjoint data situations. In order to make a comprehensive comparison, we increase the client numbers of CIFAR10 and CIFAR100 and in each round, only 10 of clients participate in the federated procedures. Besides, we also add one real federated application ISIC2019 with multiple statistical heterogeneity problems including PCDD. The exact parameters and communication rounds of all methods can be found in the Appendix C.2.

In Table 2, we divide CIFAR10 and CIFAR100 into 10, 50 and 100 clients and keep the PCDD degree in the same level. As can be seen, with the number of client increasing, the performance of all methods drops greatly. No matter in the situations of fewer or more clients, our method achieves better performance and outperforms best baseline by 2.98% in CIFAR-10 and 0.95% in CIFAR-100 on average. Besides, in Table 2, we verify FedMR with other methods under a real federated applications: ISIC2019. As shown in the last line of Table 2, our method achieves the best improvement of 3.41% relative to FedAvg and of 1.09% relative to best baseline MOON, which means our method is robust in complicated situations more than PCDD[2].

### 4.4 Further Analysis

Except performance under PCDD, we here discuss the communication cost between clients and server, local burden of clients, and privacy, and conduct the ablation study.

---

[2]Note that, in Appendix, we provide two real-world datasets to further demonstrate the efficiency of FedMR compared with baselines.

Table 5: Performance of FedMR on CIFAR10 and CIFAR100 when only 50%, 80% and 100% of clients are allowed to submit their class prototypes under different partitions.

| Datasets | Split | FedAvg | 50% | 80% | 100% |
|---|---|---|---|---|---|
| CIFAR10 | P10C3 | 77.82 | 80.42 | 80.27 | 82.75 |
| | P50C3 | 75.46 | 79.14 | 79.43 | 79.58 |
| | P100C3 | 71.65 | 76.73 | 76.87 | 76.93 |
| CIFAR100 | P10C10 | 54.31 | 56.41 | 56.73 | 57.27 |
| | P50C10 | 50.22 | 51.06 | 52.57 | 53.36 |
| | P100C10 | 47.90 | 48.80 | 48.88 | 49.60 |

Table 6: The ablation study of FedMR. We illustrate average accuracy of FedMR on the four datasets without the inter-class loss or the intra-class loss or both. Results of all partitions are shown in Appendix C.7.

| Inter | Intra | FMNIST | SVHN | CIFAR10 | CIFAR100 |
|---|---|---|---|---|---|
| - | - | 79.35 | 86.48 | 78.58 | 66.41 |
| ✓ | - | 80.18 | 87.46 | 78.86 | 66.62 |
| - | ✓ | 81.03 | 87.15 | 80.89 | 67.11 |
| ✓ | ✓ | **83.45** | **87.98** | **82.48** | **67.66** |

**Communication Concern.** In terms of communication cost, our method needs to share the additional prototypes. To show how much extra communication cost will be incurred, in Table 3, we show the number of transmission parameters in each round to compare the communication cost of methods with prototypes (Fed-Proc and FedMR) and without prototypes (FedAvg, FedProx, FedNova, MOON, FedDyn and FedDC). From the results, the additional communication cost in single round is negligible. Except for sharing class prototypes averaged from class representations, we also tried to use 1-hot vectors to save computation and communication, but the performance is unsatisfactory and even worse than the FedAvg. This is because such discriminative structure might not fit the optimal statistics as each class has different hardness to learn, and it is not clear that arriving at such an optimization stationary from the given initialization can be a good choice. In Table 4, we also provide the communication rounds on CIFAR100 under three different partitions, where all methods require to reach the best accuracy of FedAvg within 400 rounds. From the table 4, we can see that FedMR uses less communication rounds (best speedup) to reach the given accuracy, indicating that FedMR is a communication-efficient approach.

**Local Burden Concern.** In real-world federated applications, local clients might be mobile phones or other small devices. Thus, the burden of local training can be the bottleneck for clients. In Figure 4(a), we compute the number of parameters that needs to be saved in local clients and the average local computation time per round. As can be seen, FedDC, FedDyn and MOON require triple or even quadruple storing memory than FedAvg, while FedProc and FedMR only need little space to additionally store prototypes. In terms of local computation time, FedMR requires more time to carefully reshape the feature space. To handle some computing-restricted clients, we provide light versions of FedMR, namely Lite 10 and Lite 50, where local clients randomly select only 10 or 50 samples to compute inter-class loss. From Figure 4(a), the training time of Lite 10 and Lite 50 decreases sharply, while their performance is still competitive and better than other baselines, as shown in Figure 4(b). Please refer to Appendix C.6 for more details.

**Privacy Concern.** Although prototypes are population-level statistics of features instead of raw features, which are relatively safe (Mu et al. (2021); Tan et al. (2022)), it might be still hard for some clients with extreme privacy limitations. To deal with this case, one possible compromise is allowing partial local clients not to submit prototypes. In Table 5, we verify this idea for FedMR on CIFAR10 and CIFAR100, where at the beginning, we only randomly pre-select 50% and 80% clients to require prototypes. From Table 5, even under the prototypes of 50% clients, FedMR still performs better than FedAvg, showing the elastic potential of FedMR in the privacy-restricted scenarios. In the extreme case where all prototypes of clients are disallowed to be submitted, we can remove the inter-class that depends on prototypes from FedMR and use the vanilla federated learning with the intra-class loss. As shown in Table 6, it can achieve a promising improvement than that without the intra-class loss. Note that, we cannot counteract the privacy concerns of federated learning itself, and leave this in the future explorations.

Table 7: Variance of top 50 egienvalues of covariance matrices of all classes within a mini-batch on CIFAR10.

| Method | P5C2 | P10C2 | P10C3 | P10C5 | IID |
|---|---|---|---|---|---|
| FedAvg | 1201 | 1274 | 1055 | 814 | 640 |
| FedProx | 931 | 977 | 874 | 727 | 582 |
| FedProc | 1044 | 1074 | 955 | 739 | 522 |
| FedNova | 1234 | 1277 | 977 | 755 | 572 |
| MOON | 749 | 866 | 774 | 744 | 579 |
| FedDyn | 854 | 906 | 844 | 759 | 599 |
| FedDC | 1077 | 1104 | 784 | 766 | 572 |
| FedMR (intra) | 478 | 437 | 372 | 407 | 538 |
| FedMR (inter+intra) | 570 | 538 | 566 | 579 | 635 |

Table 8: The average accuracy of all methods adopted in Table 1 with or without the aid of prototypes on FMNIST. Since FedProc is the prototype version of FedAvg, here we don't show them.

| Method | FedProx | FedNova | MOON | FedDyn | FedDC | FedMR |
|---|---|---|---|---|---|---|
| w/o Prototypes | 79.89 | 75.84 | 79.87 | 80.26 | 80.09 | **81.03** |
| w Prototypes | 80.29 | 78.76 | 80.12 | 81.21 | 80.57 | **83.45** |

**Decorrelating Analysis**  Here, we empirically verify the effectiveness of FedGELA compared with all methods on constraining feature spaces from the view of the variance of eigenvalues of the covariance matrix M. In Table 7, we show the variance of top-50 eigenvalues (sort from the largest to the smallest) of the covariance matrix M within a mini-batch (batchsize is 128) after training 100 rounds on CIFAR10, calculated as: $\frac{1}{128} \sum_{i=1}^{50} (\lambda_i - \frac{1}{50} \sum_{j=1}^{50} \lambda_j)^2$. As can be seen, when the PCDD problem eased, the variance of FedAvg gradually drops, which means in more uniform data distribution, the variance should be relatively small. The slight and sharp decreasing variance of prior federated methods and our FedMR indicate that they indeed help but still suffer from the dimensional collapse problem caused by PCDD while FedMR successfully decorrelates the dimensions. However the variance under the intra-class loss is too small and far away from the values of FedAvg in the IID setting, meaning it may enlarge the risk of the space invasion. In order to prevent space invasion, our inter-class loss provide a margin for the feature space expansion. In the table, we could see that the variance of FedMR (under the intra-class loss and the inter-class loss) is a little larger compared to FedMR (the intra-class loss) and approaches to the FedAvg under the IID setting.

**Ablation Study.**  FedMR introduces two interplaying losses, the intra-class loss and the inter-class loss, to vanilla FL. To verify the individual efficiency, we conduct an ablation experiment in Table 6. As can be seen, the intra-class loss generally plays a more important role in the performance improvement of FedMR, but their combination complements each other and thus performs best than any of the single loss, confirming our intuition to prevent the collapse and space invasion under PCDD jointly. Besides, as FedMR and FedProc need local clients additionally share class prototypes which might not fair for other baselines, in Table 8, we properly configure all baselines with prototypes on FMNIST to show the superiority of FedMR.

## 5  Conclusion

In this work, we study the problem of *partially class-disjoint data* (PCDD) in federated learning, which is practical and challenging due to the unique collapse and invasion problems, and propose a novel approach called FedMR to address the dilemma of PCDD. Theoretically, we show how the proposed two interplaying losses in FedMR to prevent the collapse and guarantee the proper margin among classes. Extensive experiments show that FedMR achieves significant improvements on FedAvg under the PCDD situations and outperforms a range of state-of-the-art methods.

## Acknowledgements

The work is supported by the National Key R&D Program of China (No. 2022ZD0160702), STCSM (No. 22511106101, No. 22511105700, No. 21DZ1100100), 111 plan (No. BP0719010) and National Natural Science Foundation of China (No. 62306178). Ziqing Fan and Ruipeng Zhang were partially supported by Wu Wen Jun Honorary Doctoral Scholarship, AI Institute, Shanghai Jiao Tong University.

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
