# OpenReview forum: "Federated Learning under Partially Disjoint Data via Manifold Reshaping"
_TMLR — Accepted by TMLR_

### Review · Reviewer_9MLk · 2023-07-07

**Summary Of Contributions:**

The paper addresses the challenge of statistical heterogeneity in federated learning (FL) and introduces a novel approach called FedMR. Existing FL methods often assume access to samples from all classes during local training, neglecting the practical scenario of partially disjoint data (PDD) where clients contribute only a subset of classes. FedMR reshapes the feature space in local training by incorporating intra-class and inter-class losses. The intra-class loss prevents feature collapse, while the inter-class loss ensures appropriate margins between categories. Experimental results demonstrate that FedMR outperforms existing FL methods in terms of accuracy and communication efficiency. FedMR effectively addresses the challenges posed by PDD, providing a promising solution for FL in real-world scenarios.


**Audience:**

Yes

**Broader Impact Concerns:**

None.

**Claims And Evidence:**

Yes

**Requested Changes:**

I don't expect any major changes. It would be great to provide clarification for the points listed above in the weaknesses section.
It would be even better to quantify the notion of DPP and study the performance of the method theoretically.

**Strengths And Weaknesses:**

Strengths
------------

1. The paper introduces a novel approach called FedMR specifically designed to tackle the challenges posed by partially disjoint data (PDD) in federated learning (FL). By reshaping the feature space through the incorporation of intra-class and inter-class losses, FedMR addresses the unique collapse and invasion characteristics of PDD, which have been underexplored in previous FL methods.


2. The paper provides extensive experimental evaluations on various datasets to demonstrate the effectiveness of FedMR. The results consistently show that FedMR achieves significantly higher accuracy and improved communication efficiency compared to existing FL methods.

3. Overall the paper is nicely written with clear motivation with the help of examples. There are some things which could be improved to make it easier to follow. Please see the weaknesses part for these.






Weaknesses and questions:
-------------------------------

1.  Section 3.3.1 lacks intuition and explanations. Please provide some discussion on why uniformity of eigenvalues would prevent dimension collapse?


2.  Theorem 1. Assumes the dimensions are not correlated, which is fine to get a result. However, in the discussion the authors mention that the dimensions become uncorrelated by the use the intra-class loss. I cannot see a proof for this.  Also in practice it would probably not be possible to reach perfectly disentangled dimensions by minimizing the intra-class loss? If so, it would be better to generalize the theorem 1 to include this case.


3. I like the motivating example (3.2), I am assuming the calculations are without the FedMR. Could you also demonstrate how FedMR works in this example?

4. What is (p_k^c)^t ?
5. Does theorem 1 depend on the class distributions and overlap across the clients?
6. Why not use 1-hot vectors for the class prototypes? This could reduce communication cost.

---

### Review · Reviewer_q3Gk · 2023-07-10

**Summary Of Contributions:**

This paper introduces FedMR, a new federated learning method that improves model quality when clients have partially disjointed data (PDD) with respect to classes. PDD essentially means that some clients may not have data from all classes, and the paper claims that this would cause issues such as space invasion and mode collapse, hurting FL model quality. The solution is to use a manifold-reshaping approach to prevent degeneration caused by missing local classes. This is done by distributing global prototypes of all classes to all clients as references, establishing proper margins among classes to avoid space invasion. Tests on FMNIST, SVHN, CIFAR10, CIFAR100, and ISIC2019 show that FedMR achieves higher accuracy than other methods like FedAvg, FedPro, and FedProc.

**Audience:**

Yes

**Broader Impact Concerns:**

No ethical concerns.

**Claims And Evidence:**

Yes

**Requested Changes:**

1. In Figure 1, the authors show that their approach help avoid space invasion. However, have the authors verified that for other FL methods (e.g., FedProx, FedProc, MOON), they also do not have this space invasion issue?

2. It seems FedMR incurs more computation overhead than other approaches, where computation is the main factor that affects energy consumption on mobile/edge devices. A lite version is included, but the comparison of the lite version is only made against weak baseline FedAvg. Can the authors provide a more direct comparison to SOTA FL approaches in terms of computation overhead and accuracy?

3. Please clarify whether FedMR requires heavy tuning of hyperparameters to achieve its best result. Can you provide a more direct comparison between FedAvg (or any existing method that is more directly comparable to FedMR but is not PDD aware) and FedMR, without tuning any hyperparameters but just adding the manifold reshaping terms?

4. Please improve the writing and readability of the manuscript, e.g.,
	- The format of the references appears to be incorrect. References can be placed inside parentheses to distinguish them from the main text. "Federated learning McMahan et al. (2017); Li et al. (2020a); Yang et al. (2019) has drawn…" -> Federated learning (McMahan et al. (2017); Li et al. (2020a); Yang et al. (2019)) has drawn…".
	- "…by adding the proximal regularization on parameters in FedAvg", the first mention of FedAvg is missing a reference.
        - "This motivations a plenty of explorations to address" -> "This motivates a plenty of explorations to address"
        ...

**Strengths And Weaknesses:**

Strengths:

- The introduced scenario of partially disjointed data with respect to classes seems indeed to be more realistic, as it’s difficult to ensure that all client devices have data from all classes.

- The paper includes measurement of the performance of various existing FL methods under this PDD setup (e.g., Table 1) and demonstrates the improvements made by FedMR. The paper also provides a comprehensive analysis of different aspects of FL methods, including accuracy, computation, communication, and memory usage.

Weaknesses:

- While showing promising results, the paper does not clearly show whether space invasion is the only reason prior works achieve sub-optimal performance under the PDD setups.

- The cost analysis of the manifold reshaping terms, such as inter- and intra-class loss calculation, shows that FedMR has a much higher computation overhead than other FL methods, being about 4.5 times more expensive than FedAvg. This could be a major limitation for choosing FedMR, as FL also runs on mobile and edge devices where energy consumption is crucial. A lite version has been provided to reduce the computation overhead of FedMR, but the comparison is only made against FedAvg, not more advanced baselines like FedProx and MOON.

- Only one real dataset, ISIC2019, is used for evaluation, while the rest are toy datasets like FMNIST and CIFAR10/100.

- There is also a contradiction in the evaluation methodology. While the authors claim to use a fixed set of hyperparameters for all methods, such as a learning rate of 0.01 as described in Section 4.1, Table 10 in the appendix states that they tune hyperparameters like the learning rate from 0.000001 to 1.

- The paper seems to have some writing issues that affect readability.

---

> ### Author Response · Authors · 2023-07-31
> **Reply to Reviewer q3Gk(part 1)**
>
> **We appreciate the constructive comments of the reviewer and try our best to address the remaining concerns. The detailed responses to the weakness and requested changes are presented as below. Note that, we highlight the contents for the requested changes in blue color in the revised submission.  In the following, "W", "R" and "A" denote the shorthand of weakness, requested changes and answer respectively.**
>
> > **W1 and R1**:"While showing promising results, the paper does not clearly show whether space invasion is the only reason prior works achieve sub-optimal performance under the PDD setups.""In Figure 1, the authors show that their approach help avoid space invasion. However, have the authors verified that for other FL methods (e.g., FedProx, FedProc, MOON), they also do not have this space invasion issue?"
>
>
> **A1**: Thanks for the constructive comments. We would like to re-clarify our claim. As depicted in Figure 1(c), we have identified two phenomena: space invasion and dimensional collapse, which are both potential problems to achieve sub-optimal performance under the condition that some classes totally miss. It is important to note that other federated learning (FL) methods can also indirectly address the space invasion issue although they are not tailored for this problem. For instance, class prototypes in FedProc and global feature representations in MOON can mitigate the problem of space invasion. Additionally, methods such as FedProx and FedDyn may constrain the negative influence of PDD in the perspective of optimization. However, it is worth highlighting that prior to our work, none of these methods systematically consider this problem, and our FedMR provide a better solution to solve it supporting by the higher performance in the experiments. In the revision, we have made it more clear in the Introduction in blue color.
>
> > **W2 and R2**:"..., A lite version has been provided to reduce the computation overhead of FedMR, **but the comparison is only made against FedAvg, not more advanced baselines like FedProx and MOON**.""..., A lite version is included, **but the comparison of the lite version is only made against weak baseline FedAvg**. Can the authors provide a more direct comparison to SOTA FL approaches in terms of computation overhead and accuracy?"
>
>
> **A2**: Figure 4 in the main paper has already compared the lite version (denoted as Lite 10 and Lite 50) against all baselines on the datasets in terms of the average memory consuming, computation time and performance. Besides, Table 3 in the main paper and Table 11 in the appendix shows the concrete values of additional memory consuming, computation time and performance on all datasets of all baselines including our light versions. (no changes from the origional submits)
>
> > **W3**:"Only one real dataset, ISIC2019, is used for evaluation, while the rest are toy datasets like FMNIST and CIFAR10/100.
>
>
> **A3**: We additionally test our FedMR on two more datasets, named HyperKvasir and ODIR in Section C.3. Here are the results on HyperKvasir under three partitions:
>
> | split | FedAvg | FedProx | MOON | FedDyn | FedProc | FedMR | $\Delta$ |
> | --- | --- | --- | --- | --- | --- | --- | --- |
> | p10c2 | 69.67 | 68.43 | 69.33 | 70.02 | 69.54 | 70.79 | +0.75 |
> | p10c3 | 76.44 | 76.92 | 77.43 | 77.01 | 76.72 | 78.32 | +0.89 |
> | p10c5 | 89.48 | 89.23 | 89.13 | 89.34 | 89.27 | 89.54 | +0.06 |
>
>
> Here are the results on ODIR under three partitions:
>
> | split | FedAvg | FedProx | MOON | FedDyn | FedProc | FedMR | $\Delta$ |
> | --- | --- | --- | --- | --- | --- | --- | --- |
> | p3c3 | 59.89 | 60.24 | 59.85 | 60.78 | 60.88 | 61.79 | +0.91 |
> | p4c3 | 56.53 | 56.42 | 56.89 | 57.12 | 55.78 | 57.67 | +0.55 |
> | p5c5 | 54.51 | 54.07 | 54.11 | 54.17 | 53.73 | 54.62 | +0.21 |

---

### Review · Reviewer_dGrL · 2023-08-07

**Summary Of Contributions:**

The paper provides a new approach for Federated Learning (FD) in classification. The paper tackle the particular case of classification FD where dataset of each local client doesn't contain the same classes from a global dataset while having the same features $\mathbf{X}$. Since actual methods are poorly adapted to this setting, they propose a new approaches based on ‘‘manifold reshaping'', i.e. by adding an intra-class and inter-class regularization of the training loss in each local client training w.r.t. representations of local neural networks models.

**Audience:**

Yes

**Broader Impact Concerns:**

No.

**Claims And Evidence:**

Yes

**Requested Changes:**


- What do you use as distribution and classifier for the example in Section 3.2. Claims in this section seem coming without support/details.

- I am not completely familiar with specificity of federated learning but your example in Section 3.2 seems to be close to a multi class classification with one versus one setting. In this situation, it is difficult for me to see what are the links between FedMR and one vs one classification. Or at least, why your MR method would be advantageous compared to doing a one vs one ?

- I am not sure to understand how you construct $\hat{z}^c_{k,i}$. Each representation space is suppose to be multivariate. According to your notations, how do you compute $\sigma_{k}^{c}$, that is suppose to be a matrix? Could you please specify the space where quantity you introduce belong to?

Minor comments:

- The format of your citations are not good. For example: Line 1: use \citep{cite1,cite2,cite3} instead of (\cite{cite1,cite2,cite3}). These errors are present in the entire text, See Section 4.1 of the TMLR LaTeX stylefile to see the citation guidelines.

- Figure 1: I think it would be more appropriate to leave only 4 classes. in the examples (a) and (b) with the same colors than in (c). The fact that you want to put $n$ classes in (a) , (b) reduces the ease of comprehension.

- Where the name ‘‘prototypes'' used in Section 3.3.2 comes from? It just represents the class average representation.

**Strengths And Weaknesses:**

The paper is well written, except for the citation requirement, which does not seem to be understood by the authors (see below).

The approach is novel and tackles an interesting problem. Although expected, the experimental results are interesting and show their approach outperforms standard ones on this particular task.


Beyond PDD framework, I would like to see how the author' approach perform on non-PDD framework compare to SOTA FD approaches.
Some additional concerns are described below.

---

### Decision · Action_Editors · 2023-09-29

**Recommendation:** Accept with minor revision

**Comment:**

This paper considers a particular form of heterogeneity in federated networks, in which the aim is to solve a multi-class classification and each client only contains a subset of the classes.

All reviewers found that the work solves an interesting/relevant problem and found the empirical results compelling. One remaining concern is that it is unclear exactly why existing methods have suboptimal performance in this scenario, and in particular, whether the hypothesis that they suffer from dimensional collapse/space invasion has been effectively tested.

I tend to agree with the authors that the work is already meaningful in that it aims to systematically address issues of space invasion/collapse and the proposed modifications seem to clearly improve accuracy in the scenario of interest. However, I also agree with Reviewer q3Gk that it would be beneficial to understand to what degree prior works may inherently address these issues (e.g., using the intuition the authors provided), and to determine whether these are key underlying issues leading to poor performance in this setting. I am thus recommending that the paper be accepted with a minor revision, where I encourage the authors to submit a minor revision of the work that incorporates this feedback via a more detailed discussion, analysis, and/or a simple experiment (e.g., perhaps expanding on the results in Table 7 for more methods) for their final camera-ready.

**Audience:**

This work, which aims to address issues of heterogeneity in federated multi-class classification, would be of interest to the TMLR audience.

**Claims And Evidence:**

All reviewers found that this work aims to solve an interesting/relevant problem, and found the empirical results (around improving accuracy in settings of partially disjoint data in FL) to be clear and compelling. The reviewers had several questions regarding the methodology, which were almost all addressed during the rebuttal phase. One remaining concern is that it is unclear exactly why existing methods have suboptimal performance in this scenario, and in particular, whether the hypothesis that they suffer from dimensional collapse/space invasion has been effectively tested, which I believe could be addressed through a minor revision.